# Novel Hydroxypyridine Compound Protects Brain Cells against Ischemic Damage In Vitro and In Vivo

**DOI:** 10.3390/ijms232112953

**Published:** 2022-10-26

**Authors:** Ekaterina Blinova, Egor Turovsky, Elena Eliseikina, Alexandra Igrunkova, Elena Semeleva, Grigorii Golodnev, Rita Termulaeva, Olga Vasilkina, Sofia Skachilova, Yan Mazov, Kirill Zhandarov, Ekaterina Simakina, Konstantin Belanov, Saveliy Zalogin, Dmitrii Blinov

**Affiliations:** 1Department of Clinical Anatomy and Operative Surgery, Department of Pharmaceutics Technology and Pharmacology, Sechenov University, 8/1 Trubetzkaya Street, 119991 Moscow, Russia; 2Department of Fundamental Medicine, National Research Nuclear University MEPHI, 31, Kashirskoe Highway, 115409 Moscow, Russia; 3Institute of Cell Biophysics of the Russian Academy of Sciences, Federal Research Center “Pushchino Scientific Center for Biological Research of the Russian Academy of Sciences”, 3 Institutskaya Street, 142290 Pushchino, Russia; 4Laboratory of Pharmacology, Department of Pathology, National Research Ogarev Mordovia State University, 68 Bolshevistskaya Street, 430005 Saransk, Russia; 5Laboratory of Molecular Pharmacology and Drug Design, Department of Pharmaceutical Chemistry, All-Union Research Center for Biological Active Compounds Safety, 23 Kirova Street, 142450 Staraja Kupavna, Russia; 6Department of Pharmaceutical Technology and Pharmacology, Scientific Centre for Expert Evaluation of Medicinal Products of the Ministry of Health of the Russian Federation, 8/2 Petrovsky Blvd, 127051 Moscow, Russia

**Keywords:** experimental brain ischemia, cerebral cortex cell culture, glucose-oxygen deprivation, hydroxypyridine compound, cell death, oxidative stress, neurological disorder, gene expression

## Abstract

A non-surgical pharmacological approach to control cellular vitality and functionality during ischemic and/or reperfusion-induced phases of strokes remains extremely important. The synthesis of 2-ethyl-6-methyl-3-hydroxypyridinium gammalactone-2,3-dehydro-L-gulonate (3-EA) was performed using a topochemical reaction. The cell-protective effects of 3-EA were studied on a model of glutamate excitotoxicity (GluTox) and glucose-oxygen deprivation (OGD) in a culture of NMRI mice cortical cells. Ca^2+^ dynamics was studied using fluorescent bioimaging and a Fura-2 probe, cell viability was assessed using cytochemical staining with propidium iodide, and gene expression was assessed by a real-time polymerase chain reaction. The compound anti-ischemic efficacy in vivo was evaluated on a model of irreversible middle cerebral artery (MCA) occlusion in Sprague-Dawley male rats. Brain morphological changes and antioxidant capacity were assessed one week after the pathology onset. The severity of neurological disorder was evaluated dynamically. 3-EA suppressed cortical cell death in a dose-dependent manner under the excitotoxic effect of glutamate and ischemia/reoxygenation. Pre-incubation of cerebral cortex cells with 10–100 µM 3-EA led to significant stagnation in Ca^2+^ concentration in a cytosol ([Ca2+]i) of neurons and astrocytes suffering GluTox and OGD. Decreasing intracellular Ca^2+^ and establishing a lower [Ca2+]i baseline inhibited necrotic cell death in an acute experiment. The mechanism of 3-EA cytoprotective action involved changes in the baseline and ischemia/reoxygenation-induced expression of genes encoding anti-apoptotic proteins and proteins of the oxidative status; this led to inhibition of the late irreversible stages of apoptosis. Incubation of brain cortex cells with 3-EA induced an overexpression of the anti-apoptotic genes *BCL-2*, *STAT3*, and *SOCS3*, whereas the expression of genes regulating necrosis and inflammation (*TRAIL*, *MLKL*, *Cas-1*, *Cas-3*, *IL-1β* and *TNFa*) were suppressed. 3-EA 18.0 mg/kg intravenous daily administration for 7 days following MCA occlusion preserved rats’ cortex neuron population, decreased the severity of neurological deficit, and spared antioxidant capacity of damaged tissues. 3-EA demonstrated proven short-term anti-ischemic activity in vivo and in vitro, which can be associated with antioxidant activity and the ability to target necrotic and apoptotic death. The compound may be considered a potential neuroprotective molecule for further pre-clinical investigation.

## 1. Introduction

Strokes have been a leading cause of death and disability worldwide for decades [1]. They impose an excessively growing financial and economic burden on modern society [2]. Broad timely implementation of thrombolytic therapy and highly effective endovascular surgical interventional technologies drastically changed clinical outcomes of an acute ischemic brain injury but raised an issue of reperfusion-associated cellular and tissue damage [3]. Hence, a non-surgical pharmacological approach to control cellular vitality and functionality during ischemic/hypoxic and/or reperfusion-induced phases of stroke is of extreme importance.

Recent decades were full of impressive advantages in distinguishing cellular and tissue mechanisms of strokes. A role of glutamatergic excitotoxicity, calcium signaling, reactive oxygen species (ROS), and cellular death mechanisms is conformed in numerous of fundamental articles [4]. Alongside progress in stroke pathophysiology, novel molecules have been developed and tested as a promising way to modulate identified targets and therefore impact the consequences of the disease. Of these molecules, the ones with antioxidant activity particularly stand out. The anti-ischemic property of selenium, melatonin, natural and hemi-modified flavonoid antioxidants, and numerous synthetic agents has been demonstrated recently in different laboratory stroke-representing settings [5,6,7,8]. Hydroxypyridine derivatives have long been considered the molecules that act as membrane stabilizers to protect cellular and tissue integrity from inflammatory and hypoxic/ischemic damage [9]. Thus, in vitro and in vivo studies demonstrated the anti-ischemic activity of hydroxypyridine structures in the endothelin-1 model of experimental strokes [10]. It was then shown that derivatives with different substitutes containing, in particular, malate, L-arginine, and succinate moiety acted as powerful cell-protectants under a condition of ischemic and reperfusion heart, retinal and brain cells damage, and oxidative stress activation [11,12,13,14]. In the late-2010s, in the course of the novel compounds of hydroxypyridine origin search, an ascorbic acid-containing derivative was synthetized at the All-Union Research Center for Biological Active Compounds Safety. It was taken into consideration that the antioxidative property of vitamin C has also been well-known for decades, although its efficacy in strokes remains controversial [15]. While new evidence of its protective role has been emerging, recent systematic reviews and meta-analyses performed by Myung and co-authors find no strong evidence of a clinically meaningful intake of ascorbic acid-containing formulations in cardiovascular diseases including strokes [16]. The new compound possessed lipid-regulating and antioxidant properties in chronic rats and rabbits experiments; its LD_50_ obtained through intraperitoneal administration in mice was equal to 720 ± 11 mg/kg (patent RU 2743923 C1). As previous experimental studies of compounds of similar origin demonstrated strong evidence of their impact on brain function, the new molecule was supposed to cross the blood-brain barrier.

These results were an incentive for us to explore the brain-protective property of the novel molecule, an ascorbic acid-containing derivative of hydroxypyridine (3-EA). The main goal of our study was to assess the anti-ischemic activity of 3-EA on in vitro and in vivo model of cerebral ischemia.

## 2. Results

### 2.1. Synthesis of 3-EA

As a result of a synthesis carried out as a topochemical reaction of specially prepared samples of 2-ethyl-6-methyl-3-hydroxypyridine and gammalactone-2,3-dehydro-L-gulonic acid (ascorbic acid), 7.12 g (90.9%) crystalline water-soluble powder of 2-ethyl-6-methyl-3-hydroxypyridinium gammalactone-2,3-dehydro-L-gulonate (laboratory name–3-EA) with creamy tint was received. The pH of 3% solution is 4.5–5.5.

*2-ethyl-6-methyl-3-hydroxypyridinium gammalactone-2,3-dehydro-L-gulonate* (Figure 1). MW: 313.44. FT-IR (ν; cm^−1^ KBr): 3390 (OH), 2691 (CH), 2573 (N+), 1673 (C=C), 1578 (C=N). Mass spectrum: 314.43 (M + 1). H-NMR (400 MHz): DMSO (d6), δ, ppm CH_2_ 1.77–1.88 (8H.m), 2.67–2.87 (4H.m); C=C 7.82 (m, 5H), N+ 7.98–8.25 (m, 4H). Elemental analysis for C_14_H_19_NO_7_: Calculated,%: C 53.67; H 6.19; N 4.47; O 35.74. Found,%: C 53.62; H 6.21; N 4.43.

### 2.2. Ca^2+^ Dynamics in Murine Brain Cortex Cells Incubated with Different Concentrations of 3-EA

Cell cultures of the mouse cerebral cortex were preincubated with various concentrations of 3-EA (10, 50, and 100 μM) for 24 h. In a model of glutamate excitotoxicity (GluTox), neurons of the cerebral cortex reacted with a rapid high-amplitude increase in calcium concentration ([Ca2+]i) and followed the establishment of a new stationary level for Ca^2+^ without the ions pumping out of the cytosol throughout the entire recording time (Figure 2A, black curve). After a 24 h incubation of the cells with 10 µM 3-EA under GluTox condition, a rapid increase in [Ca2+]i was also recorded; then there was a trend towards pumping out Ca^2+^ from the cytoplasm (Figure 2A, orange curve). Increasing 3-EA concentration to 50 and 100 µM not only reduced the primary increase in [Ca2+]i in response to GluTox, but also lowered the level of [Ca2+]i at the new steady state of the Ca^2+^ signaling system compared to GluTox without 3-EA or when using 10 µM 3-EA (Figure 2A, red and blue curves). Astrocytes were less sensitive to GluTox, since the cells responded to the stimulus with a rapid increase in [Ca2+]i followed by the utilization of Ca^2+^ from the cytosol (Figure 2B). In the case of astrocytes, 3-EA showed less efficiency in suppressing Ca^2+^ signals on GluTox—the amplitudes of the primary increase in [Ca2+]i were almost identical at all 3-EA concentrations used. However, there was a trend towards a higher pumping rate of Ca^2+^ ions after GluTox in the case of preincubation with 50 and 100 μM 3-EA (Figure 2B, red and blue curves). 

### 2.3. An Impact of 3-EA on Survival of Cortex Cells under GluTox Condition

Less than 10% of cells survived after a 24 h exposure of GluTox to the cells of the cerebral cortex. Early and late stages of apoptosis were recorded in 7% and 42% of the cells. Necrotic death occurred in 50% of the cell population (Figure 3A,B, red columns), whereas in the control without glutamate exposure, cellular death was recorded in no more than 8–12% of the cells. 24 h-long incubation of the culture with 10 μM 3-EA followed by a 24-h GluTox revealed no significant increase in the number of surviving cells. However, a significant redistribution regarding the death pathways was observed due to a decrease in the rate of late stages of apoptosis (21%) and an increase in the rate of the early stages (23%) without a significant decrease in necrotic death (Figure 3A, blue symbols). Elevation of the 3-EA concentration up to 50 μM led to a decrease in the number of necrotic cells (38%) and cells at the late stages of apoptosis (35%) compared with GluTox. The culture preincubation with 100 μM 3-EA increased the percentage of viable cells to 29% and reduced the number of cells in the late stages of apoptosis (16%) and the percentage of necrotic death (18%) (Figure 3A, green symbols).

### 2.4. An Influence of 3-EA on Calcium Dynamics and Survival of Cells under Oxygen-Glucose Deprivation (OGD) Condition

Oxygen-glucose deprivation (OGD) in vitro reflects the spectrum of intracellular events occurring in the neuroglial network during ischemia, including Ca^2+^ dynamics in neurons and astrocytes [17]. Ischemia-like conditions for 40 min in vitro caused a biphasic increase in [Ca2+]i in neurons (Figure 3A) and astrocytes (Figure 4B) of the cerebral cortex, which correlated with necrotic death cells detected by the appearance of propidium iodide fluorescence in the cellular nuclei (Figure 4C, OGD). After 24 h preincubation of brain cells with various concentrations of 3-EA, dose-dependent suppression of OGD-induced [Ca2+]i growth in neurons (Figure 4A) and astrocytes (Figure 4B) was observed, with no effect on the first reversible phase of the [Ca2+]i elevation. At the same time, a decrease in the number of necrotic cells was observed after 40 min of OGD, with the most pronounced effect when using 100 μM 3-EA (Figure 4C).

A more toxic model of ischemia-like conditions, oxygen–glucose deprivation for 2 h followed by reoxygenation in a CO_2_ incubator for 24 h (OGD/R) led to necrotic death of about 70% of the cells and the rest of the cells were in the late stages of apoptosis (Figure 5 and Figure A1). Preincubation of cerebral cortex cells with 3-EA induced a dose-dependent decrease in the number of cells in the late stages of apoptosis. Simultaneous suppression of necrotic death after OGD/R was due to an increase in the percentage of viable cells and cells in the early stages of apoptosis (Figure 5 and Figure A1). The most effective concentration of 3-EA for necrosis inhibition was 100 μM.

### 2.5. Gene Expression in Cortex Cells Treated with 100 μM 3-EA under OGD/R Condition

It turned out that 100 µM 3-EA significantly reduced the basic expression of the *TRAIL*, *MLKL*, and *Cas-1* genes encoding proteins involved in the induction of necrotic cell death. At the same time, there was also an increase in the expression of the anti-apoptotic *Bcl-2*, *Stat3*, and *Socs3*, which occurred simultaneously with the downregulation of the pro-apoptotic *Bcl-xL* and the pro-inflammatory genes of IL-1β and TNFα (Figure 6A). After OGD/R, there was a change in the gene expression pattern: 3-EA promoted the downregulation of *MLKL* and *Cas-1* associated with necrosis inhibition. Although an OGD/R-induced increase in the expression of pro-apoptotic genes *BAX*, *Bcl-xL*, and *Nf-κB* was observed, there was also a two-fold or more upregulation of anti-apoptotic genes *Bcl-2*, *Stat3*, and *Socs3* accompanied by a downregulation of pro-inflammatory *Cas-3*, *IL-1β*, and *TNFα* genes (Figure 6B). Interesting results were registered among the expression of genes encoding protein regulators of the redox status. 3-EA induced a decrease in the baseline (Figure 6C) and OGD/R-induced (Figure 6D) level of expression of *Mao-A* and *Mao-B* (localized in mitochondria), and an upregulation of Catalase gene localized in the cytoplasm.

### 2.6. An Efficacy of 3-EA on Animal Model of Acute Brain Ischemia in Rats

The acute ischemic attack led to a profound depression in the neurological status of experimental animals assessed 1, 3, and 7 days after middle cerebral artery (MCA) occlusion (Figure 7A). The animals demonstrated severe motor and sensitivity dysfunction, which resulted in an increase of a neurological score to 4.6 ± 0.4, 3.8 ± 0.3, and 3.2 ± 0.4 at the three subsequent time points. 3-EA significantly prevented the severity of neurological symptoms assessed at day 3 and day 7 of the experimental pathology (*p* < 0.05; Figure 7A), while showing no impact on the earlier stage of laboratory stroke development. We registered congruous changes in the rate of damaged neurons and severity of tissue alteration (Figure 7B,D) in the studied groups: 18.0 mg/kg 3-EA daily administration for 7 days resulted in preserving neurons population in targeted area accompanied by alleviation of cerebral tissue pathological changes.

Cerebral ischemic attack elevated local oxidative stress activity (OSA) up to 7.3 ± 0.2 × 10^3^ imp./s, and simultaneously depressed antioxidant capacity to 1.0 ± 0.2 × 10^3^ imp./s (*p* = 0.001 in comparison with sham-operated animals) registered 7 days after the pathology onset (Figure 8). Experimental therapy with 3-EA decreased the magnitude of OSA activation—the index averaged at 4.9 ± 0.3 × 10^3^ imp./s (*p* = 0.04 in comparison with control group). At the same time, the compound preserved relatively high AC of cerebral tissue in the ischemic damage zone. 

## 3. Discussion

Ischemic brain damage includes a cascade of signaling and metabolic events leading to the induction of cell death via a necrotic or apoptotic process. A decrease in the oxygen partial pressure along with paucity of glucose and other nutrients supply lead to an increase in the concentration of extracellular glutamate, elevation of [Ca2+]i in neurons and astrocytes, induction of ROS production, and ultimately a change in the expression patterns of genes and proteins that regulate cell death [18,19]. It is well-known that OGD causes Ca^2+^-dependent damage to brain cells, which occurs as a result of a biphasic increase in calcium ions in the cytosol of both neurons and astrocytes. It is also generally accepted that the second phase of the global increase in [Ca2+]i is precisely one of the causes of rapid cell death resulting from impaired Ca^2+^ homeostasis [20,21]. Our experiments have shown that the preincubation of cerebral cortex cells with 3-EA in the concentration ranging from 10 to 100 µM leads to the suppression of the global increase in [Ca2+]i during OGD and GluTox, which correlates with the inhibition of cell death due to necrosis.

Restoration of blood flow after brain reoxygenation with thrombolytics is a widely used approach for the treatment of ischemic strokes. However, reperfusion therapy is limited by a narrow time window and the use of this approach beyond this optimal time can lead to additional damage to the blood-brain barrier, hemorrhagic transformation, and massive cerebral edema [22]. At the same time, it is known that reoxygenation after OGD leads to an even greater increase in ROS production, activation of inflammatory factors, disruption of Ca^2+^ homeostasis, and loss of high-energy phosphate compounds, which ultimately leads to activation of apoptosis [20]. It has been shown that 3-EA leads to the inhibition of not only necrosis, but also apoptosis. Preincubation of the cortex culture with the compound 3-EA resulted in preserving the viability of brain cells demonstrating a downregulation of *TRAIL*, *MLKL*, *Cas-1*, and *Bcl-xL* genes and an increase in *Bcl-2* gene expression. Bcl-2 family proteins are known to regulate caspase-dependent apoptosis resulting from calcium overload and mitochondrial dysfunction in brain cells [23]. Therefore vitality tests, and the gene expression profile reverberate an ability of the novel compound to prevent not only a necrotic but an apoptotic way of cell death.

Neuroinflammation significantly contributes to brain tissue damage during the primary and secondary progression of brain ischemia/reperfusion injury [24]. Therefore, the role of the transcription factors Socs3 and Stat3 in the anti-inflammatory protection of brain cells in OGD/R is of extreme importance. Current scientific opinions include evidence for both pro- and anti-apoptotic effects of these factors [25,26]. Our experiments have convincingly demonstrated that 3-EA anti-apoptotic cell-protective effect can be mediated by the overexpression of *Stat3* and *Socs3* and the downregulation of *Cas-3*, *TNT-α*, and *IL-1β* genes. 

We also assume that the modulation of the brain cortex cells’ redox status sufficiently contributes to 3-EA protective property. Monoamine oxidase A and B (Mao-A and Mao-B) degrade amines and are also a source of mitochondrial ROS [27,28]. Moreover, Mao-A is expressed at a high level in neurons, whereas Mao-B is expressed in astrocytes [29,30]. Mao inhibitors prevent dopamine-induced mPTP pore formation and cell death [31]. Catalase is a key enzyme in brain cells that utilizes ROS during ischemia and reoxygenation [32]. Its overexpression in brain cells before MCA occlusion has a powerful protective effect, while this approach is not effective in the therapeutic regimen [33]. In our experiments, the incubation of cerebral cortex cells with 3-EA lead to the suppression of basal and OGD/R-induced expression of genes encoding both forms of monoamine oxidase; the protective effect of 3-EA is observed both in neurons and astrocytes. As for the gene encoding catalase, its baseline expression increased almost threefold after pretreatment of cells with 3-EA. The increased level of catalase persisted after OGD/R. 

Our in vitro results are supported by the data obtained in animal experiments. Daily intravenous administration of 18.0 mg/kg 3-EA during a week following MCA occlusion ameliorates the severity of the neurological disorder and preserves neurons’ cellular population and brain tissue integrity. In addition, restraining the activity of local oxidative stress and simultaneous prevention of ischemia-induced AC’s drastic depression highlights the importance of the antioxidant property of the novel hydroxypyridine compound. 

We clearly comprehend some important limitations of our study. Among them, our choice to use only the model of irreversible brain ischemia without reperfusion, a relatively narrow range of biological tests carried out on the animal stroke model, and only the short-term follow-up period after MCA occlusion. The study of gene expression alone is not sufficient enough to conclude with certainty on the different modulations of intracellular homeostasis, while a study at the protein level would be more informative. At the same time, mentioned issues will be addressed in our further research work.

## 4. Materials and Methods

### 4.1. 3-EA Synthesis

The specially prepared samples of 2-ethyl-6-methyl-3-hydroxypyridine and gammalactone-2,3-dehydro-L-gulonic acid were dried at 100–105 °C to a constant weight then ground to obtain particles no larger than 50 μm in size. Then, 3.43 g (0.025 gmol) of 2-ethyl-6-methyl-3-hydroxypyridine and 4.4 g (0.025 gmol) of gamma-lactone-2,3-dehydro-L-gulonic acid were charged into the homogenizer and the mass was then homogenized for 2 h at a shift speed of 300–400 rpm. The received product was analyzed to control the substance’s structure and purity. 

### 4.2. Cell Culture Preparation

Cell co-cultures of cortical neurons and astrocytes were isolated from the brains of newborn NMRI mice. The cortex was excised with clippers, put in a test tube, incubated for 2 min and the supernatant was removed with a pipette. The cells were then covered with 2 mL trypsin (1% in Ca^2+^- and Mg^2+^-free Versene solution) and incubated for 10 min at 37 °C under constant shaking at 600 rpm. Trypsin was then deactivated by an equal volume of cold embryo serum, and the cells were washed twice with Neurobasal A medium before being resuspended in Neurobasal medium containing glutamine (0.5 mM) B-27 (2%) and gentamicin (20 μg/mL). 200 μL of the suspension was put in a glass ring (internal diameter of 6 mm) resting on a round 25 mm coverslip (VWR International) coated with polyethyleneimine (one hippocampus for ten glasses). The glass ring was removed after a 5 h incubation period in a CO_2_ incubator (37 °C) and the culture medium was added to the dishes. Approximately 2/3 of the volume of the culture medium was replaced every 3 days. The density of plated cells was 15.000 cells/sq cm and the age of the cell cultures was 10 days in vitro (DIV). One part of dishes with cultures from the same passage was used in experiments for the measurements of cytosolic Ca^2+^ concentration ([Ca2+]i) and vitality tests. The remaining dishes with cell cultures were used for the extraction of total RNA and further real-time polymerase chain reaction (RT-PCR) assay.

### 4.3. Modelling of Ischemia-like Conditions

Ischemia-like conditions (OGD) were achieved by omitting glucose (HBSS medium without glucose) and by displacement of dissolved oxygen with argon in the leak-proof system [34]. The level of oxygen in the medium was measured using a Clark electrode. Oxygen tensions reached values of 30–40 mm Hg or less within 20 min after the beginning of displacement. Ischemia-like conditions lasting for 40 min were created by supplying the OGD medium into the chamber with cultured cortical cells. Constant argon feed into the experimental chamber was used to prevent the contact of the OGD medium with the atmospheric air. 

For experiments to identify the role of 3-EA on the induction of necrosis and apoptosis, OGD conditions were created for 2 h. For this, the culture medium was changed to Neurobasal A-medium without glucose and pH was balanced 7.35–7.4 by adding 10 mM HEPES. The displacement of oxygen was carried out in a closed chamber using argon blowing. The chamber with Petri dishes was in a thermostat at 37 °C and 95% humidity. After 2 h OGD, the cell cultures were removed from the thermostat, 0.2 mM D-glucose was added to each Petri dish, and the cells were returned to the CO2 incubator for reoxygenation for 24 h [31].

### 4.4. Simulation of Glutamate Excitotoxicity

To create acute (~50 min) conditions for the excitotoxic action of glutamate (GluTox), 100 mM glutamate was added to the cells in a magnesium-free medium with the addition of 20 μM glycine. The effects of GluTox on neurons and astrocytes were evaluated by measuring the amplitude, shape, and rate of cell calcium response and assessment of cell viability before and after GluTox conditions.

To induce long-term GluTox and reveal the role of 3-EA on apoptosis processes, 100 μM glutamate plus 20 μM glycine was added to the culture medium for 24 h, while magnesium ions were present in the culture medium.

### 4.5. Measurement of Cytosolic Calcium Concentration

For recordings of changes in [Ca2+]i cortical cell cultures were loaded with Fura-2AM (4 µM; 40 min incubation; 37 °C). Cells were stained with the probe dissolved in Hank’s balanced salt solution (HBSS) composed of (mM): 156 NaCl, 3 KCl, 2 MgSO_4_, 1.25 KH_2_PO_4_, 2 CaCl_2_, 10 glucose, and 10 HEPES, pH 7.4. After incubation with the dyes, cells were washed three times before the experiment. To measure the [Ca2+]i, we used the system based on the inverted motorized microscope Leica DMI6000B (Leica, Germany) with a high-speed monochrome CCD-camera Hamamatsu C9100 (Japan) and a high-speed light filter replacing system Leica’s Ultra-Fast Filter Wheels with replacing time 10–30 ms. For excitation and registration of Fura-2 fluorescence, we used the FU-2 filter set (Leica, Germany) with excitation filters BP340/30 and BP387/15, beam splitter FT-410 and emission filter BP510/84, objective Leica HC PL APO 20/0.7 IMM, and excitation light source Leica EL6000 with a high-pressure mercury lamp HBO 103 W/2. 

In order to identify neurons KCl application was used. To identify the glial cells, especially astrocytes, we used a short-term ATP application. We determined the amplitudes and shape of Ca^2+^ responses under oxygen-glucose deprivation (OGD) and glutamate excitotoxicity (GluTox). 

### 4.6. Assessment of Cell Viability and Apoptosis

Cell death induced by OGD, reoxygenation (R), or glutamate excitotoxicity exposure was assessed by propidium iodide (PI, 1 µM) before and after treatment in the same microscopic field. It is known that viable cells are not permeable to PI, while Hoechst 33342 easily penetrates through the plasma membrane, staining the chromatin. According to a commonly used method [17], hippocampal neurons were defined as apoptotic if the intensity of Hoechst fluorescence was 3–4 times higher compared to Hoechst fluorescence in healthy cells, indicating chromatin condensation which can occur as a result of apoptosis induction. The fluorescence of the probes was registered with a fluorescent system on the basis of an inverted fluorescent microscope Axio Observer Z1 equipped with a high-speed monochrome CCD-camera Hamamatsu ORCA-Flash 2.8. The Lambda DG-4 Plus illuminator (Sutter Instruments, Novato, CA, USA) was used as a source of the excitation of fluorescence. The fluorescence of the probes was detected with an inverted fluorescent microscope Zeiss Axio Observer Z1 using Filter Set 01 and Filter Set 20. Five different fields were analyzed for each coverslip. Each experiment was repeated three times using separate cultures.

We used different times of cell cultures exposure to OGD and GluTox for the purpose of assessment of ischemia and reoxygenation input to the necrotic and apoptotic processes of cerebral cortex cell death. 

### 4.7. Extraction of RNA

Mag Jet RNA Kit (Thermo Fisher Scientific, Waltham, MA, USA) was used for the extraction of total RNA. The RNA quality was estimated by electrophoresis in the presence of 1 μg/mL ethidium bromide (2% agarose gel in Tris/Borate/EDTA buffer). The concentration of the extracted RNA was determined with a NanoDrop 1000c spectrophotometer. RevertAid H Minus First Strand cDNA Synthesis Kit (Thermo Fisher Scientific, USA) was used for reverse transcription of total RNA.

### 4.8. Real-Time Polymerase Chain Reaction (RT-qPCR)

Each PCR was performed in a 25 μL mixture composed of 5 μL of qPCRmix-HS SYBR (Evrogen, Moscow, Russia), 1 μL (0.2 μM) of the primer solution, 17 μL water (RNase-free), and 1 μL cDNA. Dtlite Real-Time PCR System (DNA-technology, Moscow, Russia) was used for amplification. The amplification process consisted of the initial 5 min denaturation at 95 °C, 40 cycles of 30 s denaturation at 95 °C, 20 s annealing at 60–62 °C, and 20 s extension step at 72 °C. The final extension was performed for 10 min at 72 °C. All the sequences were designed with FAST PCR 5.4 and NCBI Primer-BLAST software and all the primers were synthesized by Evrogen (Moscow, Russia) (Table A1). The data were analyzed with Dtlite software (DNA-technology, Moscow, Russia). The studied genes’ expression was normalized to gene encoding glyceraldehyde 3-phosphate dehydrogenase (GAPDH). The data were analyzed using Livak’s method [35].

### 4.9. Laboratory Animals

Sprague-Dawley male rates and NMRI mice were purchased at the Pathogen-free Laboratory Animals Breeding Facility of Shemiakin and Ovchinnikov Institute of bioorganic chemistry of Russian Academy of Sciences (Pushchino, Russia). Animal study protocol containing particularities of experiments in rats and murine brain cell culture development was reviewed and approved by Sechenov University Local Ethic Committee on February 03, 2022 (Rev. No 03/2022/12-4). Animals were kept in separate cages under natural daylight conditions with free access to food and water. Air temperature (20–22 °C) and 55–60% humidity were maintained. 

### 4.10. Experimental Stroke Modelling

Cerebral ischemia in Sprague-Dawley male rats weighing 200–220 g was achieved using a middle cerebral artery (MCA) occlusion model by [36] in our modification in rats. In brief, a rat was anesthetized with 5% isoflurane (Aerrane, Baxter, Deerfield, IL, USA) in 30% O_2_/70% N_2_O using the V-10 Anesthesia system (VetEquip Inc., Pleasanton, CA, USA). Following the induction of anesthesia, the level of isoflurane was reduced and maintained at 1.5%. The animal was positioned supine, its head was fixed in a stereotaxic system, and a midline neck skin incision was performed. After that, the dissection was carried between the left sternohyoid muscle and the left sternomastoid muscles until the carotid sheath is observed. The common carotid artery (CCA) was then thoroughly dissected from the internal jugular vein and the vagus nerve from the level of sternum up to the bifurcation. The superior thyroid artery and the ascending pharyngeal artery–branches of the external carotid artery were dissected and coagulated. Vascular micro-clamps were placed on the CCA proximally and on the internal carotid artery (ICA) distally. Then the ECA was ligated with a 10-0 suture distally and transected. Two loose loops of 10-0 suture were placed on the ECA stump and a 7-0 suture with a rounded tip was inserted into the ECA stump’s lumen. The loose loops were then tightened, the micro-clamps were removed, and the 7-0 suture with a rounded tip was drawn intraluminally 200 mm cranially to occlude the MCA. After hemostasis was ensured, the wound was closed. A decrease in cerebral blood flow was followed and confirmed by laser flowmetry (Biopac MP160, Biopac Systems Inc., Goleta, CA, USA). 

### 4.11. Experimental Intervention and Assessments

All animals were randomly allocated into three groups, six rats in each one. Animals in a group of experimental intervention were administered intravenous (IV) 3-EA 18 mg/kg in 0.5 mL sterile saline 5 min after experimental cerebral ischemia disorder’s onset and then daily for 6 days at the same time, while the shame-operated rats and control animals with MCA occlusion received an equivalent volume of saline alone in the same regimen. The neurological disorder was measured 1, 3, and 7 days after MCA occlusion by the modified Bederson scale [37]. On day 7 all animals were euthanized and their brains were extracted and dissected. Injured areas were separated into two relatively equal parts, one of which was then fixed in a 10% phosphate-based formalin solution and the other was processed for biochemical testing.

### 4.12. Histological Examination

Four-μm-thick sections of the formalin-fixed and paraffin-embedded brain tissue samples were stained with hematoxylin and eosin (BioVitrum, Moscow, Russia) and 0.1% toluidine blue (BioVitrum, Moscow, Russia) using the Nissle method. Leica DM4000 B LED microscope, equipped with a Leica DFC7000 T digital camera running by the LAS V4.8 software (Leica Microsystems, Wetzlar, Germany) was used for the examination of the sections. The measurements of the damaged areas were made using Leica Application Suite, version 4.9.0 (Leica Microsystems, Wetzlar, Germany). To objectify the results of the histological examination, we used the semi-quantitative analysis (scoring) of slides stained by the Nissle. The signs of cerebral tissue degeneration (pericellular and perivascular edema, hyperchromic shriveled neurons, irregularly shaped neurons, hypochromic neurons, satellitosis, neuronophagy, and pyknotic neurons) in every slide were evaluated with 0 to 3 according to the scoring table (Table A2). The analysis of cortex tissue damage was performed on the rats’ brain parietal lobe at the side of MCA ligation. The areas of ischemic brain alteration were assessed in five random fields at ×200 magnification.

### 4.13. Antioxidant Capacity Assessment

We assessed the antioxidant capacity (AC) of brain tissues by Fe-activated chemiluminescence method on day 7 of experimental stroke modeling [38,39]. Damaged lobes of animals’ brains were excised, rinsed in cold saline, and then homogenized with 0.5 mL PBS per 100 mg of cerebral tissue on ice (0–4 °C). Having centrifuged the homogenate at 3000× *g* for 15 min, 0.3 mL of supernatant was poured into a cuvette containing 0.5 mL of phosphate buffer (pH 7.5). Base chemiluminescence was measured by SmartLum-1200 (DISoft, Russia) after adding 0.02 mL of 2% hydrogenic peroxide to the cuvette. Activated by 1 mL of 0.05 mM Fe(II) sulfate chemiluminescence was registered for 5 min. 

### 4.14. Data Handling and Statistical Analysis

In vitro data were obtained from at least three coverslips and three different cell preparations. Image J, Origin 8.5, and Prism GraphPad (La Jolla, CA, USA) software were used in order to analyze data, create graphs, and perform statistical tests. All values are given as mean ± SD or as responses of individual neurons. Data were analyzed to assess normality of distribution and then statistically compared using a paired *t*-test; they were considered significant at *p* ≤ 0.001. Animal data were also presented as mean ± SD. Time-dependent intergroup comparisons were made using ANOVA followed by performing post-hoc the Tukey’s test, while the significance of the difference between experimental and control groups on day 3 was confirmed with independent two-tail *t*-test using STATA 17.0 software (Stata Corp. LLC, College Station, TX, USA).

## 5. Conclusions

3-EA suppressed cortical cell death in a dose-dependent manner under the excitotoxic effect of glutamate and ischemia/reoxygenation. Pre-incubation of cerebral cortex cells with 10–100 µM 3-EA led to significant stagnation in Ca^2+^ concentration in a cytosol ([Ca2+]i) of neurons and astrocytes suffering GluTox and OGD. Decreasing intracellular Ca^2+^ and establishing a lower [Ca2+]i baseline inhibited necrotic cell death in an acute experiment. The mechanism of 3-EA cytoprotective action involved changes in the baseline and ischemia/reoxygenation-induced expression of genes encoding anti-apoptotic proteins and proteins of the oxidative status, which led to inhibition of the late irreversible stages of apoptosis. Incubation of brain cortex cells with 3-EA induced an overexpression of the anti-apoptotic genes *BCL-2*, *STAT3*, and *SOCS3*, whereas the expression of genes regulating necrosis and inflammation (*TRAIL*, *MLKL*, *Cas-1*, *Cas-3*, *IL-1β*, and *TNFa*) were suppressed. The administration of 3-EA 18.0 mg/kg intravenously and daily for 7 days following MCA occlusion preserved rats’ cortex neuron population, decreased the severity of neurological deficit, and maintained the antioxidant capacity of damaged tissues. Thus, 3-EA demonstrated proven short-term anti-ischemic activity in vivo and in vitro, which can be associated with antioxidant activity and the ability to target necrotic and apoptotic death. The compound may be considered as a potential neuroprotective molecule for further pre-clinical investigation.

## Figures and Tables

**Figure 1 ijms-23-12953-f001:**
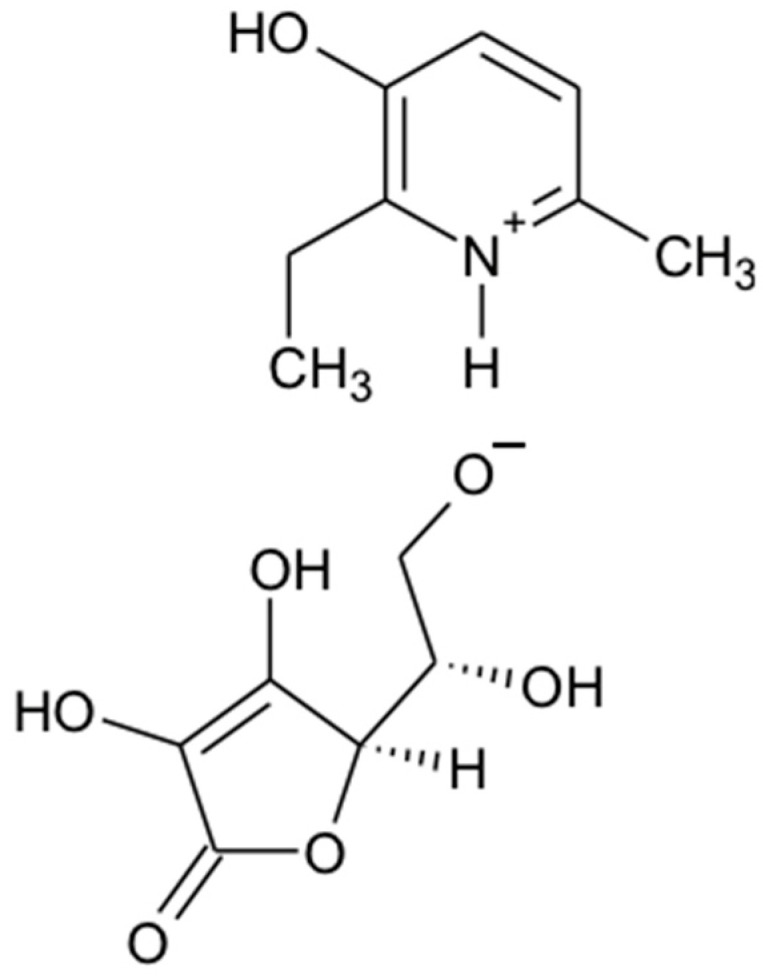
Chemical structure of 2-ethyl-6-methyl-3-hydroxypyridinium gammalactone-2,3-dehydro-L-gulonate (3-EA).

**Figure 2 ijms-23-12953-f002:**
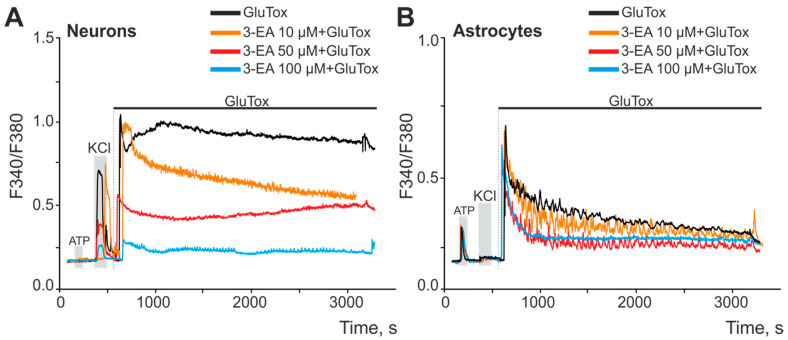
An effect of 24 h preincubation of cortical cells with various concentrations of 3-EA on the increase in [Ca2+]i upon induction of glutamate excitotoxicity for 50 min (GluTox, 100 μM glutamate + 20 μM glycine in Mg^2+^-free medium). (**A**,**B**)—averaged Ca^2+^ signals of neurons (**A**) and astrocytes (**B**) of the cerebral cortex during the induction of glutamate excitotoxicity. Short-term (30 s) applications of KCl (35 mM) and ATP (10 µM) were performed to detect neurons and astrocytes, respectively. The experiments were performed in triplicate on three cell cultures of different passages.

**Figure 3 ijms-23-12953-f003:**
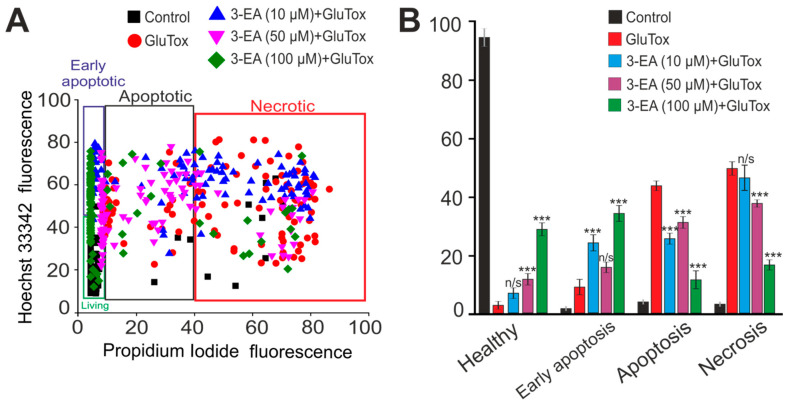
An effect of 24 h preincubation of the cortex cells with various concentrations of 3-EA on the viability of the cells during 24 h glutamate excitotoxicity (GluTox): (**A**) Cytogram showing the viability of cerebral cortex cells (horizontal axis–propidium iodide fluorescence intensity; vertical axis–Hoechst 33342 fluorescence intensity). Cells were stained with probes 24 h after GluTox outbreak. (**B**) An effect of 24 h preincubation of the cortex cells with various concentrations of 3-EA on the induction of necrosis and apoptosis with GluTox: the results are presented as Mean ± SD. Number of cell cultures = 3; number of coverslips with cells for each sample = 5; n/s-data are unreliable (*p* > 0.05), *** *p* < 0.001 when comparing experimental groups with GluTox (ANOVA and Tukey’s post-hoc test).

**Figure 4 ijms-23-12953-f004:**
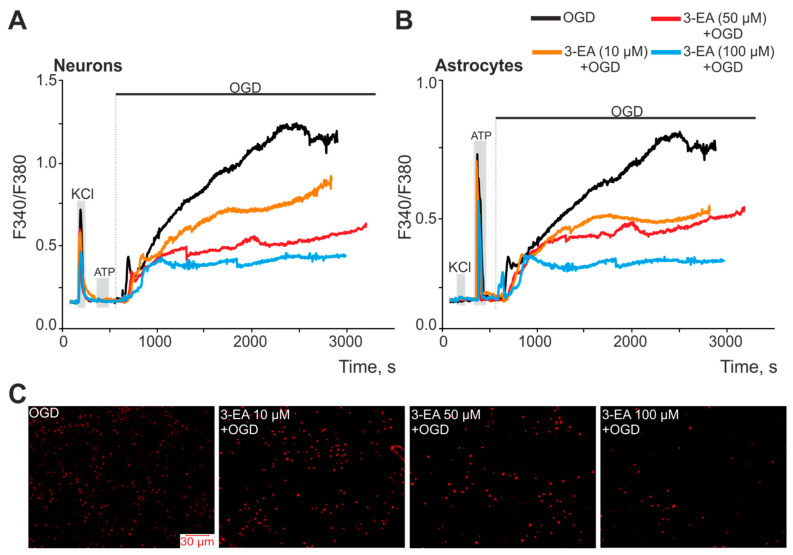
An effect of 24 h preincubation of cerebral cortex cells with various concentrations of 3-EA on the increase in [Ca2+]i during 40 min of OGD: an average Ca^2+^ signals of neurons (**A**) and astrocytes (**B**) of the cells during OGD. Short-term (30 s) applications of KCl (35 mM) and ATP (10 µM) were performed to detect neurons and astrocytes, respectively. (**C**) Staining of the cell culture with the propidium iodide fluorescent probe, reflecting necrotic death after 40 min of OGD. The experiments were performed in triplicate on three cell cultures of different passages.

**Figure 5 ijms-23-12953-f005:**
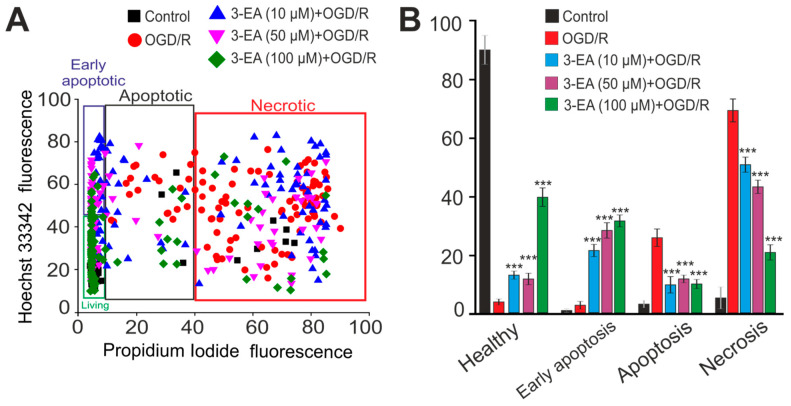
An effect of 24 h preincubation of the cortex cells with various concentrations of 3-EA on the viability of the cells during 2 h oxygen-glucose deprivation with subsequent reoxygenation during 24 h (OGD/R): (**A**) Cytogram showing the viability of cerebral cortex cells (horizontal axis–propidium iodide fluorescence intensity; vertical axis–Hoechst 33342 fluorescence intensity). Cells were stained with probes after 24 h of reoxygenation. (**B**) An effect of 24-h preincubation of the cortex cells with various concentrations of 3-EA on the induction of necrosis and apoptosis with OGD/R: The results are presented as mean ± SD, number of cell cultures = 3; number of coverslips with cells for each sample = 5; n/s-data are unreliable (*p* > 0.05), *** *p* < 0.001 when comparing experimental groups with GluTox (ANOVA and Tukey’s post-hoc test).

**Figure 6 ijms-23-12953-f006:**
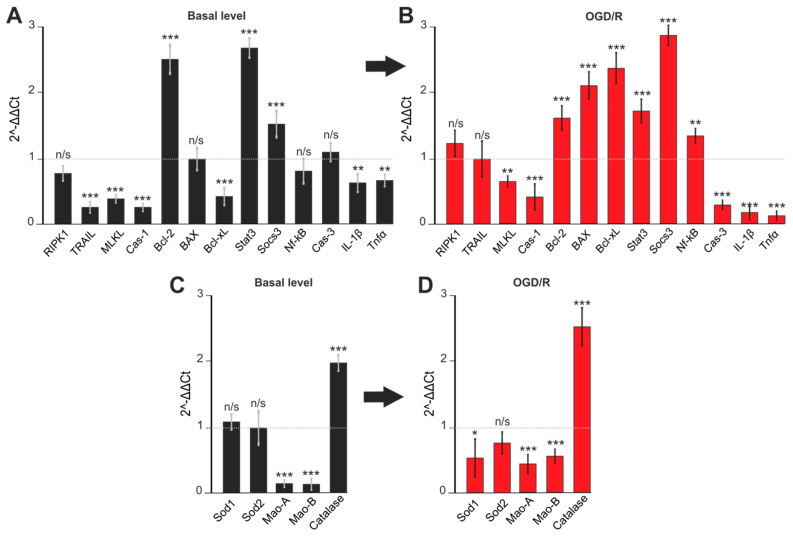
Expression of genes regulating apoptosis (**A**,**C**) and oxidative stress activity (**B**,**D**). Gene expression in control cells are marked by dashed line for basal level. Gene expression in OGD/R cells are marked by dashed line for OGD/R level. Comparison of experimental groups regarding control or OGD/R group without 3-EA: n/s-data not significant (*p* > 0.05), * *p* < 0.05, ** *p* < 0.01 and *** *p* < 0.001 when comparing experimental groups (ANOVA and Tukey’s post-hoc test). Comparison of experimental groups relative to each other is indicated in red or black. The number of RNA samples is 3. The number of animals used for cell culture preparation is 3.

**Figure 7 ijms-23-12953-f007:**
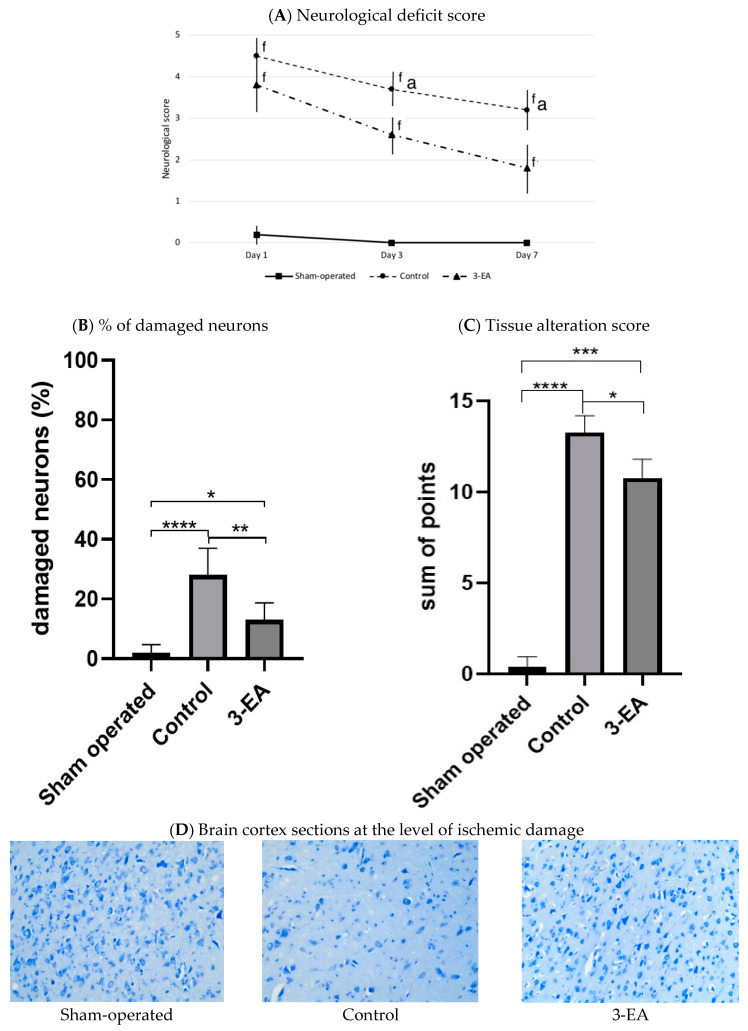
Neurological disorder and brain tissue damage in experimental groups 7 days after MCA occlusion; staining by the Nissle method, ×200; n = 6 in each group, data presented as mean ± SD, ^f^ *p* < 0.05 when compared to sham-operated group, ^a^ *p* < 0.05 when compared to control group; * *p* < 0.05, ** *p* < 0.01, *** *p* < 0.005, **** *p* < 0.001 in intergroup comparison; significance of differences was assessed by ANOVA and the Tukey’s test.

**Figure 8 ijms-23-12953-f008:**
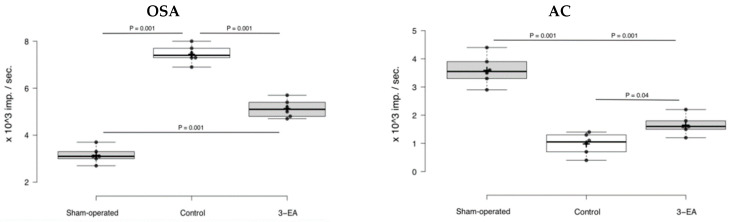
Oxidative stress activity (**OSA**) and antioxidant capacity (**AC**) of rats’ brain tissue in ischemic (for control and 3-EA groups) and anatomically related (for sham-operated group) areas; n = 6 in each group, data presented as boxplots with mean ± SD, and significance of differences was assessed by ANOVA and the Tukey’s test.

## Data Availability

The 3-EA acute toxicity information mentioned in the introduction section of the manuscript has been taken from a publicly available patent RU 2743923 C1, date of priority 29.05.2020 (In Russ). This data can be found here: https://fips.ru/registers-doc-view/fips_servlet (accessed on 28 September 2022).

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
