# Peer review of "Novel Hydroxypyridine Compound Protects Brain Cells against Ischemic Damage In Vitro and In Vivo"

_ijms, 2022, doi:10.3390/ijms232112953_

Round 1
Reviewer 1 Report
This article entitled Novel Hydroxypyridine Compound Protects Brain Cells Against Ischemic Damage In Vitro and In Vivo, by Blinova et al. studied the protective effect of 3-EA, in the context of stroke. This study was realized in vitro as well as in vivo. If the study as a whole seems well conducted, a number of points could be improved:
Major points:
· In the introduction, ascorbic acid-containing derivative of hydroxypyridine as only described as a novel molecule. This description is not enough for understanding of the interest of the study. Could the authors justify the choice of this molecule in a more concrete way (?), explain less succinctly what type of molecule it is, what source does it derive from (?), why was it chosen/designated, what are the characteristics of its toxicity (?), does it cross the BBB (What evidence ?).
· 2.1. It seems that this part is more about materials and methods than results.
· L 117. Could the others define “Glutox”?
· L 132-133 & 145-146: “No cell death was observed after the GluTox modeling experiments.” & “Less than 10% of cells survived after a 24-hour exposure of GluTox to the cells of the cerebral cortex.” These two sentences seem contradictory. Could the authors clarify to remove the doubt?
· L 147: “Early and late stages of apoptosis were recorded in 7 and 42% of the cells respectively, and necrotic death occurred in 50% of the cell population”: the term “respectively” is not clear.
· Gene expression: the study of gene expression alone is not sufficient to conclude with certainty on the different modulations. A study at the protein level would be more informative. Authors should emphasize this point in the endpoints.
· Fig 6. C: pictures of brain damage should be added to illustrate the data.
· Concerning the in vivo study, the data concern only short term study (up to 7 days post-MCAO). The absence of a long-term study have to be highlighted as a limitation to the study and the conclusions have to be moderate.
· L 485: “On day 4 all animals were euthanized”: this sentence is not right, if not, how do the authors quantify brain damage 7 days after MCA? Thanks for rectifying.
Minor pints: L 71 : "ROC" should be replaced by ROS.

Author Response
Dear Reviewer,
On behalf of all the manuscript authors I thank you for valuable remarks, which we have used as effective guide to improve our article.
Point 1.
In the introduction, ascorbic acid-containing derivative of hydroxypyridine as only described as a novel molecule. This description is not enough for understanding of the interest of the study. Could the authors justify the choice of this molecule in a more concrete way (?), explain less succinctly what type of molecule it is, what source does it derive from (?), why was it chosen/designated, what are the characteristics of its toxicity (?), does it cross the BBB (What evidence ?).
Reply
We have tried to clarify all mentioned issued in rewritten paragraph two of the Introduction.
Point 2.
2.1. It seems that this part is more about materials and methods than results.
Reply
We have carefully rereviewed the section and made it more appropriate to “result” form.
Point 3.
L 117. Could the others define “Glutox”?
Reply
We have defined “GluTox” in the section.
Point 4.
L 132-133 & 145-146: “No cell death was observed after the GluTox modeling experiments.” & “Less than 10% of cells survived after a 24-hour exposure of GluTox to the cells of the cerebral cortex.” These two sentences seem contradictory. Could the authors clarify to remove the doubt?
Reply
We have rewrite this section, the source of contradiction was technical error.
Point 5.
L 147: “Early and late stages of apoptosis were recorded in 7 and 42% of the cells respectively, and necrotic death occurred in 50% of the cell population”: the term “respectively” is not clear.
Reply
We agree with you, “respectively” has been removed from the text.
Point 6.
Gene expression: the study of gene expression alone is not sufficient to conclude with certainty on the different modulations. A study at the protein level would be more informative. Authors should emphasize this point in the endpoints.
Reply
We have emphasized this point in the limitations subsection od the “Discussion”
Point 7.
Fig 6. C: pictures of brain damage should be added to illustrate the data.
Reply
We have added brain cortex sections at the level of ischemic damage stained by the Nissle method.
Point 8.
Concerning the in vivo study, the data concern only short term study (up to 7 days post-MCAO). The absence of a long-term study have to be highlighted as a limitation to the study and the conclusions have to be moderate.
Reply
We fully agree with you, and added appropriate sentence in limitations; conclusions have also been rewritten.
Point 9.
L 485: “On day 4 all animals were euthanized”: this sentence is not right, if not, how do the authors quantify brain damage 7 days after MCA? Thanks for rectifying.
Reply
We have clarified the point, the source of contradiction was technical error.
Point 10.
Minor pints: L 71 : "ROC" should be replaced by ROS.
Reply
We have replaced ROC by ROS
Thank you very much,
Authors

Reviewer 2 Report
Authors aimed to explore protective effects of novel molecule (3-EA) using in vitro and in vivo model of cerebral ischemia. Using adequate methods they demonstrated anti-ischemic activity of this newly synthesized molecule. Still, some clarification are needed.
- 'It was then shown that derivatives with different substitutes containing in particular magnesium, malate and succinate moiety acted as powerful cell-protectants under a condition of glucose-oxygen deprivation and oxidative stress activation (11)' Is this reference valid for such claim? Please replace it with some more appropriate reference.
- Why the Authors measured Ca2+ influx in 50min GluTox and 40min OGD and viability on 24h GluTox and 2h OGD? It should be indicated in the M&M section.
- In the Appendix A Authors presented coronal sections of rat’s brain. Why didn't they use TTC staining to show infarct area? What should we see from regular brain sections?
- It would be more convenient to describe in vitro models first and then methods used.
- Authors perform Histological examination and AC of cerebral tissue, however they did not indicate which region they analyzed but only stated 'damaged areas'. Also 'The analysis was performed at the areas of alteration in 5 random fields' - what precisely 5 random fields means? Brain regions?
Minor comments:
- ' Tukay's test' instead Tukey's.
- 'Neurological disorder was measured on 1, 3 and 7 days after MCAO....On day 4 all animals were euthanized. ' Is it typographical error?
- '... performed by Myung and co-authors find no strong evidence...(12, 13)' Ref 13 is Myung et al, Ref 12 is Tang et al.
Author Response
Dear Reviewer,
On behalf of all the manuscript authors I thank you for valuable remarks, which we have used as effective guide to improve our article.
Point 1.
- 'It was then shown that derivatives with different substitutes containing in particular magnesium, malate and succinate moiety acted as powerful cell-protectants under a condition of glucose-oxygen deprivation and oxidative stress activation (11)' Is this reference valid for such claim? Please replace it with some more appropriate reference.
Reply
We have tried to clarify all mentioned issued in the rewritten Introduction.
Point 2.
- Why the Authors measured Ca2+ influx in 50min GluTox and 40min OGD and viability on 24h GluTox and 2h OGD? It should be indicated in the M&M section.
Reply
These timing in experimental modelling reflexes acute and postponed mechanism of cellular damage. Ca2+ influx is a reactive reply of neuron to OGD and GluTox, and it launches majorly necrotic mechanisms of cell death, measured 2h after OGD. At the same time, in such settings there is no enough time to assess expression of apoptotic genes. That’s why we used 24h postpone. We have indicated the reason in M&M section of the manuscript.
Point 3.
In the Appendix A Authors presented coronal sections of rat’s brain. Why didn't they use TTC staining to show infarct area? What should we see from regular brain sections?
Reply
We have removed fresh coronal brain sections because TTC staining was not used to assess damaged areas.
Point 4.
It would be more convenient to describe in vitro models first and then methods used.
Reply
We have rearranged M&M section as recommended by the reviewer.
Point 5.
Authors perform Histological examination and AC of cerebral tissue, however they did not indicate which region they analyzed but only stated 'damaged areas'. Also 'The analysis was performed at the areas of alteration in 5 random fields' - what precisely 5 random fields means? Brain regions?
Reply
We have clarified the point in M&M section of the manuscript. In particular we examined damaged areas in rat’s brain parietal lobe at the side of MCA ligation.
Point 6.
- ' Tukay's test' instead Tukey's.
Reply
We have corrected the name.
Point 7.
'Neurological disorder was measured on 1, 3 and 7 days after MCAO....On day 4 all animals were euthanized. ' Is it typographical error?
Reply
We have had appropriate amendments in the text. It was typographical error.
Point 8.
'... performed by Myung and co-authors find no strong evidence...(12, 13)' Ref 13 is Myung et al, Ref 12 is Tang et al.
Reply
We have corrected citations’ numeration in the text.
Thank you very much,
Authors

Round 2
Reviewer 1 Report
All comments have been taken into account and the scientific article has been much improved.